# Heating Capacity and Biocompatibility of Hybrid Nanoparticles for Magnetic Hyperthermia Treatment

**DOI:** 10.3390/ijms25010493

**Published:** 2023-12-29

**Authors:** Aline Alexandrina Gomes, Thalita Marcolan Valverde, Vagner de Oliveira Machado, Emanueli do Nascimento da Silva, Daniele Alves Fagundes, Fernanda de Paula Oliveira, Erico Tadeu Fraga Freitas, José Domingos Ardisson, José Maria da Fonte Ferreira, Junnia Alvarenga de Carvalho Oliveira, Eliza Rocha Gomes, Caio Fabrini Rodrigues, Alfredo Miranda de Goes, Rosana Zacarias Domingues, Ângela Leão Andrade

**Affiliations:** 1Departamento de Química, Instituto de Ciências Exatas e Biológicas (ICEB), Universidade Federal de Ouro Preto (UFOP), Ouro Preto 35400-000, MG, Brazil; alinealexandrina26@gmail.com (A.A.G.); vagnerolmach@yandex.com (V.d.O.M.); emanueli.silva@ufop.edu.br (E.d.N.d.S.); 2Departamento de Morfologia, Instituto de Ciências Biológicas (ICB), Universidade Federal de Minas Gerais (UFMG), Belo Horizonte 31270-901, MG, Brazil; thalitamarcolan@gmail.com (T.M.V.); fabrinir.caio@gmail.com (C.F.R.); 3Laboratório de Física Aplicada, Centro de Desenvolvimento da Tecnologia Nuclear (CDTN/CNEN), Belo Horizonte 31270-901, MG, Brazil; daniele.fagundes@cdtn.br (D.A.F.); fernandadepaulaoliveira46@gmail.com (F.d.P.O.); jdr@cdtn.br (J.D.A.); 4Materials Science and Engineering, Michigan Technological University, Houghton, MI 49931-1295, USA; efragafr@mtu.edu; 5Departamento de Engenharia de Materiais e Cerâmica (CICECO), Universidade de Aveiro (UA), 3810193 Aveiro, Portugal; jmf@ua.pt; 6Departamento de Microbiologia, Instituto de Ciências Biológicas (ICB), Universidade Federal de Minas Gerais (UFMG), Belo Horizonte 31270-901, MG, Brazil; junnia@outlook.com; 7Departamento de Produtos Farmacêuticos, Faculdade de Farmácia, Universidade Federal de Minas Gerais (UFMG), Belo Horizonte 31270-901, MG, Brazil; elizarochagomes@gmail.com; 8Departamento de Patologia Geral, Instituto de Ciências Biológicas (ICB), Universidade Federal de Minas Gerais (UFMG), Belo Horizonte 31270-901, MG, Brazil; alfredomgoes@gmail.com; 9Departamento de Química, Instituto de Ciências Exatas (ICEx), Universidade Federal de Minas Gerais (UFMG), Belo Horizonte 31270-901, MG, Brazil; dominguesrz@gmail.com

**Keywords:** hybrid nanoparticles, cytotoxicity, hyperthermia

## Abstract

Cancer is one of the deadliest diseases worldwide and has been responsible for millions of deaths. However, developing a satisfactory smart multifunctional material combining different strategies to kill cancer cells poses a challenge. This work aims at filling this gap by developing a composite material for cancer treatment through hyperthermia and drug release. With this purpose, magnetic nanoparticles were coated with a polymer matrix consisting of poly (L-co-D,L lactic acid-co-trimethylene carbonate) and a poly(ethylene oxide)–poly(propylene oxide)–poly(ethylene oxide) triblock copolymer. High-resolution transmission electron microscopy and selected area electron diffraction confirmed magnetite to be the only iron oxide in the sample. Cytotoxicity and heat release assays on the hybrid nanoparticles were performed here for the first time. The heat induction results indicate that these new magnetic hybrid nanoparticles are capable of increasing the temperature by more than 5 °C, the minimal temperature rise required for being effectively used in hyperthermia treatments. The biocompatibility assays conducted under different concentrations, in the presence and in the absence of an external alternating current magnetic field, did not reveal any cytotoxicity. Therefore, the overall results indicate that the investigated hybrid nanoparticles have a great potential to be used as carrier systems for cancer treatment by hyperthermia.

## 1. Introduction

Although conventional treatments such as chemotherapy and radiation therapy show good results, minimizing side effects remains a challenge. Hyperthermia is an alternative technique that aims to destroy tumors through heat. However, magnetic hyperthermia treatment only weakens tumor cells, requiring a second treatment to fully destroy cancer cells. In cancer therapy, the major difficulty is to destroy tumor cells without harming the surrounding normal tissues. One of the ways to get around this problem is protecting magnetic nanoparticles (MNPs) with polymeric coatings to prevent their agglomeration while conferring to the nanocomposite system drug release capabilities [1]. On the other hand, MNPs can ensure the mobility of the polymeric nanoparticles when a magnetic field is applied, enabling their use for drug release and for treatment by hyperthermia. A large number of strategies has been proposed in the literature to encapsulate MNPs in polymeric coatings, including emulsion polymerization [2], inverse mini-emulsion polymerization [3], and direct mini-emulsion polymerization [4]. In this work, MNPs were encapsulated by the so-called solvent displacement method (SDM) firstly developed by Fessi et al. in 1998 [5], which involves a nanoprecipitation and an interfacial deposition. Two phases, organic and aqueous, are needed for particle encapsulation. Basically, the organic phase is composed of a polymer (with or without the drug) while the aqueous phase includes water and stabilizing agents (synthetic or natural). After adding the organic phase to the aqueous phase, submicron particles can be formed spontaneously upon organic solvent evaporation under reduced pressure [6].

After the encapsulation, the MNPs needed to be tested for their biocompatibility. The hemolytic testing of biomedical materials has been used to measure hemocompatibility. Erythrocytes undergo repeated cycles of deformation during circulation in the body but they must have the capacity to resist hemolysis and fragmentation, referred to as stable erythrocytes, which maintain their morphological and whose functional characteristics are preserved. Exposure of erythrocytes to hyperthermia, or supra-optimal temperature, can lead to a decrease in erythrocyte stability resulting in hemolysis and fragmentation [7]. However, to the best of our knowledge, although many studies have been conducted to analyze the hemolysis of erythrocytes in contact with MNPs, no hemolytic investigation under an alternating current magnetic field has been conducted so far. This approach is exploited here.

The selection of the poly(L-co-D,L lactic acid) (PLDLA) was due to its favorable features of resorbability and biocompatibility. Further, the previous literature reports related to the terpolymer based on poly(L-co-D,L lactic acid-co-trimethylene carbonate) (PLD-LA-TMC) materials in skin tissue regeneration [8] and tendons [9] demonstrated the biocompatibility of this terpolymer and its versatility in a wide range of medical applications. Furthermore, TMC shows flexibility and degradation rates consistent with bone tissue engineering applications [10]. Studies have also shown that PLDLA-TMC scaffolds with uneven asymmetric pores provided favorable environments for cell proliferation and migration [11]. The PEO–PPO–PEO is a polymer known as a poloxamer. Poloxamers are US FDA-approved polymers and appear listed in the US and European Pharmacopoeia as non-toxic and non-irritant excipients, being used as solubilizers, emulsifiers, and stabilizers in a variety of pharmaceutical dosage forms, including oral, parenteral, topical, ocular, rectal, and vaginal [12]. The most widely used poloxamers in pharmaceutical preparations are poloxamer 407 (Pluronic^®^ F127, BASF) and poloxamer 188, due to their unique characteristics, including a high solubilizing capacity, low toxicity, good drug release, and compatibility with numerous biomolecules and chemical excipients [11]. In aqueous solutions, the amphiphilic nature of these copolymers results in a self-aggregation into nanosized micelles with a hydrophobic inner core and a hydrophilic outer shell. Therefore, considering the biocompatibility and bioreabsorbability of poly(L-co-D,L acid lactic-co-trimethylene carbonate (PLDLA-co-TMC) and of poly(ethylene oxide)–poly(propylene oxide)–poly(ethylene oxide) (PEO–PPO–PEO), our research group has reported on the preparation of PLDLA-co-TMC/PEO–PPO–PEO-based nanocomposites incorporating the finasteride drug prepared through the SDM method. Such a composite revealed to be promising as a drug delivery system [1]. On the other hand, hybrid nanoparticles consisting of MNP cores and PLDLA-co-TMC and PEO–PPO–PEO triblock copolymer shells in different ratios were recently investigated [13,14]. Their extensive physical–chemical characterization enabled electing the most suitable MNP/polymer ratio.

Despite these polymers being biocompatible, they are also likely to induce the formation of acidic byproducts which might lead to inflammatory reactions in tissues and clinical failure [15]. Since this novel composite system is targeted for cancer treatment by hyperthermia and drug delivery, it is essential to undertake biocompatibility and heat and drug release studies.

Therefore, the goals of the present work are the following: (i) adopting a factorial design of experiments to find the conditions that produce the smallest magnetite particle sizes; (ii) using the smallest magnetite particle sizes to prepare hybrid nanoparticles with the best coating amount obtained in previous works; (iii) evaluating the hyperthermia heating capacity; and (iv) assessing the biocompatibility and hemocompatibility by cytotoxicity and hemolytic tests in the presence and absence of an external alternating current (AC) magnetic field.

## 2. Results and Discussion

### 2.1. Determination of the Optimal Synthesis Conditions to Obtain the Smallest Particle Sizes

A previous work conducted by our research group on coating magnetic nanoparticles with varying amounts of PLDLA-co-TMC and PEO–PPO–PEO revealed that the hybrid composite particles similar to the ones prepared in the current study had hydrodynamic diameters of approximately 193 nm. Knowing that particles with hydrodynamic diameters larger than 200 nm are sequestered in the spleen by phagocytosis, it is advisable to work with the smallest possible size of magnetic nanoparticles to ensure that the prepared composite ones are smaller than 200 nm. So, the effects of addition time of the ammonium hydroxide solution, the mixing speed, and the temperature of the synthesis on the mean particle size were studied by an analysis of the factorial design and size measurements, by TEM.

#### Analysis of the Factorial Design

To obtain a synthesis condition that presented mag nanoparticles as small as possible, a 2^3^ factorial design was performed as described in Table 1. The coded levels of independent factors (addition time of the ammonium hydroxide solution, X_1_; the mixing speed, X_2_; and the temperature of the synthesis, X_3_), the second-order contrast coefficients, as well as the average nanoparticle size (Y, nm) for each experiment run are shown in Table 2.

In order to evaluate the influence of factors (X_1_; X_2_; X_3_) and the interactions between factors influencing mag sample size, analysis of variance (ANOVA) was performed with a (α) of 0.05. Figure 1 and Table 3 show the pareto chart, the effects and probability values for each factor, and the second-order interaction analyzed.

Observing the results of the effects and the *p*-values, it can be stated that the X_1_ and X_3_ main effects, as well as the X_1_X_3_ interaction effect had a significant influence on the nanoparticle size. All significant effects present positive values. Apparently, temperature is the most influencing factor and its increase leads to an increase in the size of the nanoparticles. It is important to highlight that there is a very important interaction effect between X_1_ and X_3_. In the experiments from 1 to 4 (where the temperature was at the low level), it is possible to notice that the high level of X_1_ led to smaller nanoparticles (4.3 and 4.8 nm at the low level against 3.9 and 3.2 nm at the high level of X_1_). On the other hand, in the experiments from 5 to 8 (where the temperature was at the high level), the opposite occurred, i.e., the high level of X_1_ led to particles almost twice as big (4.8 and 4.7 nm at the low level against 9.6 and 9.2 nm at the high level of X_1_). In general, particle size increases with increasing reaction time. The greater growth can be explained via different growth processes such as diffusion-limited growth, aggregation, or Ostwald ripening [16]. The kinetics of such processes are accelerated with increasing reaction temperatures [17]. These conditions were found in experiments 6 and 8, where the longer addition time of NH_4_OH resulted in a longer reaction time.

Several authors have studied the effects of different conditions on the synthesis of MNPs. Mahdavi et al. [18], for example, studied the separate effects of pH variation, temperature, and stirring rate on particle size. The temperature study showed that with an increase of the reaction temperature from 25 to 45 °C, the crystallite size was reduced. However, the crystallite size was increased from 8.3 nm to 13.2 nm when the temperature increased from 45 to 85 °C. A plausible explanation for this, according to the authors, is that by increasing the reaction temperature, there is more energy within the solution that can increase mobility and cause a greater number of collisions between particles [19]. Saragi et al. [20] varied temperature in a range between 25 °C and 80 °C in order to investigate its effect on the size and morphology of synthesized nanoparticles. They found that the average size of Fe_3_O_4_ nanoparticles increased with the synthesis temperature due to the acceleration of the chemical reaction of Fe^2+^ and Fe^3+^ ions.

Regarding the base addition speed, Gnanaprakash et al. [21] showed that the nanocrystal size decreased with an increasing alkali addition rate. They explained that when the addition rate increased, the number of nucleated particles also increased and hence, the size of the particles decreased. However, all these authors varied only one experimental parameter, not being able to verify their interaction effects, as in the case of our work.

Therefore, despite the positive effect of the addition time of the ammonium hydroxide solution, the conditions of run 4 (X_1_ = 6 min 24 s, X_2_ = 50 rpm, and X_3_ = 25 °C) were chosen for the synthesis of the mag sample, considering the obtainment of the targeted smaller particles.

The mathematical model *Y* = 5.236 + 0.923 *X*_1_ + 1.518 *X*_3_ + 1.424 *X*_1_*X*_3_ was found by statistical analysis, using coefficients associated with significant effects. In addition, response surfaces were built (Figure 2), in which it was possible to observe the experimental conditions corresponding to the desired smallest MNP sizes.

Figure 2 shows the response surfaces predicted by the mathematical model. The colors indicate the sizes of the nanoparticles: the more the color moves towards the green region on the surface, the smaller are the nanoparticles obtained under the experimental conditions indicated on the horizontal axes. The hollow dots represent the conditions of the runs indicated in Table 1 and Table 2.

### 2.2. Characterization of the Synthesized MNPs

HRTEM images of the hybrid sample (mag1@poly) are shown in Figure 3a, in addition to their respective particle size distributions and histograms (Figure 3b). The particle size distribution was performed based on bright-field TEM images utilizing FIJI/ImageJ 1.52p free software (Wayne Rasband, National Institute of Health, USA). The diameters of approximately 70 particles were manually measured to obtain their size distribution. The image allowed for inferences to be made about the shape, size, and uniformity of the particles after being chemically functionalized. The MNPs were found to be spherical/cuboidal.

The HRTEM (Figure 3a) and selected electron diffraction (SAD) data (Figure 4a) confirm magnetite to be the only iron oxide in the sample. The SAD data were further analyzed using JEMS© software (v. 3.4922U2010). For the sake of comparison, simulations were performed for the electron diffraction profiles for standard magnetite (a = 8.3967 Å) and maghemite (a = 8.33 Å), both with a centric setting space group, *Fd-3 m (Figure 4b). For such simulations, an acceleration voltage of 200 kV, a 6 nm-sized crystal, and a Lorentzian model for line shape were chosen. The results agree with the standard magnetite sample used as a reference.

Besides the dynamic effects, the magnetite phase was confirmed by confronting the experimental intensities and the simulation data for magnetite and maghemite (Figure 4b). Particularly, the (111) reflection peak for magnetite was more intense than for maghemite.

### 2.3. Stability

The measured average particle size, polydispersity index, and zeta potential data are reported in the Table 4. It can be seen that the hydrodynamic particle diameters obtained through DLS analysis are notably larger than those of the primary particles extracted from the corresponding TEM images and plotted in the histogram shown in Figure 3. This can be understood considering that the TEM images depict the primary particle sizes (in dried form), whereas the DLS data provide the hydrodynamic diameters of particles/agglomerates dispersed in a liquid, encompassing their hard cores, polymer shells, and their associations induced by magnetic, ionic, and solvent interactions [22,23]. Such interactions resulted in larger particle diameters, endowing the hybrid nanoparticles with enhanced drug delivery efficiency. Based on these considerations, the data reported in Table 4 show that the average size of the hybrid mag1@poly nanoparticles is appropriate for the intended purposes. Moreover, the PDI value is less than 0.3, indicating narrow and relatively homogenous particle size distributions, which could be regarded as a monodispersed population of particles [24].

The hydrodynamic size variations in the hybrid nanoparticles/agglomerates throughout seven days after dispersing different concentrations in water are plotted in Figure 5. The results shown in Figure 5a,b reveal that the hydrodynamic sizes and PDI values remained relatively stable for any given concentration. The sample mag1@poly at a concentration of 0.25 mg mL^−1^ was the most monodispersed and stable.

### 2.4. Heating Capability Studies

The temperature rise induced by the applied AC magnetic field on the mag1@poly nanoparticles dispersed at different concentrations, in water and silicone wax, was measured every 5 min over a time period of 60 min. The measurements were repeated under the same AC magnetic conditions using pure water and pure silicone wax in order to determine and subtract the contributions of dispersing media on the temperature rise. The hyperthermic effects due to the aqueous- and silicone wax-based ferrofluids are shown in Figure 6a and 6b, respectively. It can be clearly observed that the rate of temperature increases was strongly concentration-dependent in both media. The more concentrated was the suspension, the higher was the temperature increase. The results also show that for MNP concentrations ranging from 1 to 5 mg mL^−1^, the temperature increase was enough to reach 5 °C, a minimal rise in temperature to be used for the treatment of hyperthermia. All of these samples were able to increase the temperature of the medium in less than 5 min, evidencing a rapid hyperthermic effect. This result was independent of the media. However, the heat generation capacity of the same nanoparticles embedded in silicone wax (Figure 6b) was significantly hampered, mainly due to the hindered Brownian motion and stirring motion. These results show the influence of dispersing media in the transition between different relaxation mechanisms [25]. The relationship between the Brownian relaxation time and viscosity is given by Equation (1):(1)$$\tau_{B}\approx \frac{3\cdot V_{hydr}\cdot \eta }{k_{B}\cdot T}$$
where $V_{hydr}$ is the hydrodynamic volume of the particle, η is the viscosity of the liquid medium, $k_{B}$ is the Boltzmann constant, and T is the absolute temperature. Notice that the hydrodynamic diameter of the coated particles was larger than that of the core ones.

The effective relaxation time occurring via the two simultaneous mechanisms, Néel and Brownian, follows Equation (2) [26]:(2)$$\frac{1}{\tau_{\mathrm{eff}}}=\frac{1}{\tau_{N}}+\frac{1}{\tau_{B}}$$
where $\tau_{\mathrm{eff}}$ is the effective relaxation time, $\tau_{N}$ is the Néel relaxation time, and $\tau_{B}$ is the Brownian relaxation time.

Under an alternating current (AC) magnetic field, an MNP will rotate to line up its magnetic dipolar moment to the direction of the magnetic field. The friction of the particle with the solvent will produce heat by Brownian relaxation, which competes with the Néel relaxation that is independent of the physical particle’s rotation, both contributing to the total heating. The Brownian relaxation time increases with increasing both the hydrodynamic radii of the nanoparticle and the solvent viscosity. Nevertheless, if the $\tau_{B}$ becomes too large, $\tau_{\mathrm{eff}}$ = $\tau_{N}$ and the Néel relaxation is the sole operating mechanism responsible for the overall heat loss. Consequently, for highly viscous liquids, the Brownian component diminishes or is virtually blocked and just Néel relaxation occurs. This causes the increase in temperature of the silicone grease particles to be lower than the increase in temperature of the particles dispersed in water. This fact is observed in Figure 6a,b.

According to these results, coated MNPs can be potentially effective for localized hyperthermia treatment because it is only necessary to inject a small number of nanoparticles inside the body; besides that, the short time of application of the magnetic field will avoid the potential appearance of side effects in the patient.

### 2.5. Effect of Mag1@poly on Cellular Growth and Hemolysis with or without Magnetic Field Exposure

The evaluation of in vitro cytotoxicity is important to analyze the effects of different biomaterials’ “properties on cells” viability. Results of studies have shown high reproducibility and the costs of in vitro assays are lower when compared to tests involving animals. Here, HUVEC cells were evaluated by the MTT assay. These cells were selected for this study because they are involved in the formation of blood vessels, being widely used in studies that aim to observe impacts on angiogenesis and toxicity in blood vessels [27,28]. In addition, endothelial cells play an important role in tumor formation, inflammation, and cardiovascular disease, which are target areas for research on magnetic nanoparticle applications.

Figure 7 shows the results of the MTT assay for the samples after 24 and 48 h of exposure to mag1@poly suspensions at different concentrations. After 24 h, the results indicate that cell viability was maintained at high levels, above 70% for all mag1@poly concentrations for cells not subjected to magnetic hyperthermia (Figure 7a). This observed high viability is probably related to the fact that the coating is biocompatible. While pluronics have attracted attention for use in drug delivery systems because of their biocompatibility [29] and long blood circulation time [30,31], PLDLA-co-TMC is starting to be studied for use as a scaffold in tissue engineering due to its biological reabsorption abilities and biocompatibility.

However, after 48 h we observed a reduction in cell viability, with the concentration of 2.5 mg mL^−1^ showing the lowest value, at a limit of 70%. This can be explained by the data reported in Table 4. The samples with the largest particle sizes also exhibited the lowest viability. So, probably, the 48 h time was sufficient for particle sedimentation to occur, leading to cell death. This was also verified by another author [32]. Wu et al. (2010) [33] demonstrated that HUVEC cells exposed to different doses of magnetic nanoparticles coated with citric acid or dextran showed low cell viability, endocytosis, followed by apoptosis, in addition to significant disorganization in the cytoskeletal network [33]. The reason behind the observed cytotoxicity with dextran-magnetite was attributed by other authors to the breakdown of the dextran shell, exposing the cellular components to chains or aggregates of iron oxide nanoparticles [34].

The most significant drop in cell viability was observed for the groups exposed to the mag1@poly and submitted to the magnetic field for 30 min (Figure 7b). Moreover, the effects of magnetic hyperthermia on the cells’ toxicity were more dramatic at higher doses (2.5 and 5 mg mL^−1^) at the two analyzed time points (Figure 7b). A possible explanation for this result is the movement of the coated nanoparticles driven by the applied magnetic field favoring contact with the plasmatic membrane of the cells. Consequently, a higher rate of endocytosis may have occurred, culminating in an increase in autophagy rates and/or an increase in the amount of reactive oxygen species (ROS), causing cell death [35]. Another explanation is that cell viability decreased due to the increase in temperature caused by magnetite nanoparticles subjected to an alternating current magnetic field. Hyperthermia treatment is suitable for inducing apoptosis depending on the temperature. Hyperthermia induces changes in the fluidity and stability of the cell membrane and in the membrane’s potential. It also disrupts transmembrane transport (by altering apoptosis resistance proteins, leading to apoptosis), hinders the proper synthesis of proteins and the denaturation of the same, induces the synthesis of thermal shock proteins, damages the synthesis of RNA and DNA and inhibits the enzymes responsible for their repair, and alters the conformation of DNA [36]. Therefore, cells heated above 41 °C can begin to show signs of apoptosis and then necrosis [37]. In other words, an increase in temperature above 4 °C is sufficient to cause apoptosis and necrosis.

When comparing the results of the temperature reached in the cultivation medium at concentrations of 1 mg mL^−1^ to 5 mg mL^−1^ with the temperature previously reached by the nanoparticles in water and silicone wax (Figure 6), it was possible to observe that the temperature obtained in the culture medium (HUVEC cells) was an intermediate temperature to the two previously mentioned media. This result is due to the intermediate viscosity of the cultivation medium, as also observed in the literature, which confirms that magnetic heating and efficiency are significantly reduced when MNPs are located inside cells or tissues [38]. Another observation that reinforces what was discussed previously in this work (Figure 6b) is the influence between the Brownian relaxation time and the viscosity of the medium in which the magnetic nanoparticles were inserted.

The hemocompatibility of nanoparticles is generally assessed by a hemolytic assay. Figure 8 shows the rise in temperature of different concentrations of the hybrid nanoparticles in contact with erythrocytes. For the tested concentration range, all the hemolytic percentages were not significantly affected by the application of an AC magnetic field to the suspensions remaining within the permissible limit of 2%. According to ASTMF-756 [39], a hemolytic percentage above 2 indicates damage to the red blood cells and the presence of free hemoglobin in the bloodstream, leading to a potential health risk. Furthermore, it was possible to reach hyperthermic temperatures, i.e., an increase of the medium temperature of approximately 5 °C with all concentrations tested. Our data show the erythrocytic compatibility of the mag1@poly at all concentrations tested in samples submitted or not to magnetic hyperthermia. Future experiments will be necessary to demonstrate the interaction of nanoparticles with erythrocytes in relation to possible changes in the morphology and function of these cells.

## 3. Materials and Methods

### 3.1. Reagents

For the synthesis of magnetic nanoparticles, the following reagents were employed (high purity grade): iron (III) chloride hexahydrate (Sigma-Aldrich, Riedel-de Haen, Søborg, Denmark); sodium sulfite (Sigma-Aldrich, Riedel-de Haen, Søborg, Denmark); ammonium hydroxide (Fluka, Seelze, Germany); aqueous tetramethylammonium pentahydrate hydroxide solution at 25% (TMAOH, Sigma-Aldrich, Søborg, Germany); and hydrochloric acid (Sigma-Aldrich, Riedel-de Haen, Søborg, Denmark). For the encapsulation of the MNPs functionalized with TMAOH, the following reagents were employed: acetone and methyl alcohol (Tedia^®^ High Purity Solvents, Rio de Janeiro, Brazil); Kolliphor^®^P188 (poly(ethylene oxide) (PEO)–poly (propylene oxide) (PPO)–poly(ethylene oxide) (PEO)) triblock copolymer, MW 7680–9510 g mol^−1^ (Sigma-Aldrich, Riedel-de Haen, Søborg, Denmark); and 70:30 PLDLA-co-TMC (poly(L-co-D,L acid lactic-co-trimethylene carbonate), MW 95.820 g mol^−1^) [40]. For the biological assays, the reagents used were the following: Dulbecco’s Modified Eagle Medium (DMEM; Gibco™); Dulbecco’s Modified Eagle Medium, Nutrient Mixture F-12 (DMEM-F12; Gibco™); phosphate-buffered saline (Sigma-Aldrich, Riedel-de Haen, Søborg, Denmark); Gibco™ trypsin-EDTA (0.05%), phenol red, and 3-(4,5-dimethylthiazol-2-yl)-2,5-diphenyl-2H-tetrazolium bromide (MTT; Thermo Fisher Scientific^®^, Waltham, Massachusetts, EUA). Interventionary studies involving animals or humans, and other studies that require ethical approval, must list the authority that provided approval and the corresponding ethical approval code.

### 3.2. Methods and Techniques

#### 3.2.1. Analysis of the Factorial Design

The MNPs were prepared by co-precipitation [13,14]. Briefly, 0.5 mol L^−1^ solutions of sodium sulfite (10 mmol) and ferric chloride hexahydrate (30 mmol) were mixed in a beaker under mechanical stirring of approximately 800 rpm until the color of the resulting solution changed from dark yellow to light yellow. Then, a dilute ammonium hydroxide solution was rapidly added. After 30 min, the obtained suspension was centrifuged and the resulting wet cake (~2 g) was labeled as “mag”. The wet precipitate was then dispersed in 2 mL of TMAOH aqueous solution under stirring to obtain a stabilized ferrofluid. An aliquot of 2 mL of the stabilized ferrofluid was them mixed with 18 mL of distilled water to obtain a homogeneous suspension labeled as “mag1”, which was used as the source of core nanoparticles for the coating experiments with polymeric shells.

#### 3.2.2. Design of Mag Synthesis Experiments

To determine the optimal synthesis conditions that allowed for the formation of MNPs to be as small as possible, a 2^3^ full factorial design of experiments with a central point was implemented, considering as independent factors the addition time of the ammonium hydroxide solution (X_1_), the mixing speed (X_2_), and the temperature (X_3_). Table 1 shows the levels of the independent factors defined during the aforementioned planning. Experiments were designed in every possible combination of the levels (−1, 0, and +1) in a total of 11 experiments (triplicate at the central point). The experimental runs were performed randomly and the response (dependent variable) was the nanoparticle size (obtained by transmission electron microscopy). Data were statistically processed by employing Statistica 7.0 software.

#### 3.2.3. Coating of MNPs with PLDLA-co-TMC and PEO–PPO–PEO by SDM

The MNP cores were coated with polymeric shells by SDM to obtain hybrid nanocomposite samples. For this, an aqueous phase containing 37.5 mg of PEO–PPO–PEO dissolved in 7.00 mL of Milli-Q water was firstly prepared. An aliquot of 0.50 mL of mag1 was then added to the aqueous phase to form a total volume of 7.50 mL. The organic phase was separately prepared by admixing 20 mg of PLDLA-co-TMC to a 2.2 mL acetone + 0.3 mL methanol solution. The organic phase was added dropwise to the aqueous phase under high sonication (amplitude of 98%). The high shear was maintained for 10 min in a beaker immersed in an ice bath to avoid increasing the temperature during sonication. The colloidal suspension was then roto-evaporated to ensure effective evaporation of the organic co-solvents. Subsequently, the hybrid nanoparticles were transferred to glass vials, frozen in liquid nitrogen, and lyophilized. The lyophilized powder was stored at room temperature (25 °C) before analysis. This hybrid sample was labeled “mag1@poly”. Control shell nanoparticles (without magnetic cores), defined as “blank nanoparticles” at the bottom of Table 4, were prepared using the same experimental procedure.

#### 3.2.4. Characterization

The total amount of Fe in each sample was determined by volumetric titration with K_2_Cr_2_O_7_ [41]. High-resolution TEM (HRTEM) images and selected area electron diffraction (SAD) were collected using a thermionic-source (LaB6) transmission electron microscope, Tecnai G2-20 SuperTwin (FEI), operated at 200 kV, coupled with a Gatan Imaging Filter (GIF). The image treatment and analysis were performed using the FIJI/ImageJ 1.52p free software (Wayne Rasband, National Institute of Health, USA).

Dynamic light scattering (DLS) analysis and zeta potential (ZP) analysis were performed using a Zetasizer Nano ZS instrument (Malvern Instruments, Grovewood Rd, Malvern WR14 1XZ, UK) with a laser light wavelength of 532 nm, using diluted colloidal suspensions in filtered deionized water. The measurements were conducted at 25 ± 2 °C and the light scattering was detected at 173°. At least three measurements were obtained for each nanoconjugate system and the average and standard deviation (SD) values were calculated. The measurements were obtained at room temperature.

#### 3.2.5. Heating Ability Studies

Heat dissipation experiments were carried out by transferring each suspension with mag1@poly samples dispersed in water or in silicone grease to a test tube. The test tubes were then placed at the center of a Nova Star 5 kW RF three-loop coil instrument, with an Ameritherm power supply, consisting of a heating station induction atmosphere. The concentrations studied were 0.5, 1.0, 2.5, and 5.0 mg mL^−1^ (by magnetite). The temperature of the magnetic suspension was monitored with an optical fiber thermometer. Results were taken as the means of triplicate measurements.

### 3.3. Biological Assay

Samples from the same batch were dried in an oven at 37 °C for 48 h and used for biological assays with or without hyperthermia. After drying, the samples were subjected to UV radiation for 30 min for complete sterilization. Then, the samples were dispersed in 1× sterile PBS and gently homogenized. The obtained suspension was subsequently added in the proportions of interest to a cell culture medium to perform the in vitro tests.

#### 3.3.1. Culture of HUVEC

Human umbilical endothelial cells (HUVEC-ATCCs) were cultivated in medium with DMEM (47%; Gibco, Waltham, MA, USA), DMEM-F12 (47%; Gibco, Waltham, MA, USA), FBS (5% *v*/*v*; Gibco, Waltham, MA, USA), and penicillin/streptomycin (1% *v*/*v*; Invitrogen, Waltham, MA, USA). Cells were incubated in a humidified atmosphere of CO_2_ 5% at 37 °C and detached from the plate for the experiments at 90% confluence using trypsin-EDTA (0.25%; Gibco, Waltham, MA, USA) [27].

#### 3.3.2. Biocompatibility/Cell Cytotoxicity Assay with or without an AC Magnetic Field

The biocompatibility of mag1@poly was assessed via a 3-(4,5-dimethylthiazol-2-yl)-2,5-diphenyl-2H-tetrazolium bromide (MTT) (Invitrogen, Waltham, MA, USA) assay, as described elsewhere [42]. The MTT assay quantifies the mitochondrial activity based on the reduction of a tetrazolium salt to formazan dyes in live cells. In this study, HUVEC cells were exposed to the mag1@poly suspension, subjected or not to an AC magnetic field, and then cytotoxicity was assessed for 24 and 48 h. In summary, for the group not subjected to the magnetic field, 1 × 10^4^ HUVEC cells per well were seeded in a 48-well plate (Sarstedt, Nümbrecht, Germany), and the cells were treated with the mag1@poly suspension at final concentrations of 0.5, 1.0, 2.5, and 5.0 mg mL^−1^ diluted in 1× PBS, in addition to the viability control that received only 1× PBS and the cytotoxicity control that was exposed to 0.5% *v*/*v* Triton™ X-100 (Sigma-Aldrich, Riedel-de Haen, Søborg, Denmark) for 15 min. For the experiment with an AC magnetic field, the cell groups were prepared following the same methodology described above, including the respective viability and cytotoxicity controls, placed in sterile 50 mL plastic tubes, and subjected to the AC magnetic field of 216 kHz for 30 min. During this period, the temperature was measured and recorded every 5 min. Afterwards, the cells were plated in a 48-well plate and cytotoxicity was assessed. After the treatment period, the medium was then removed and a solution containing 130 μL of DMEM (Gibco, Waltham, MA, USA) and 100 μL of MTT (5 mg mL^−1^) was added per well. After 2 h, formazan crystals were visualized in an optical microscope and then dissolved in 130 μL of sodium dodecyl sulfate (SDS) at 10% in HCl 0.01 mol L^−1^ (Sigma-Aldrich, Riedel-de Haen, Søborg, Denmark). For all the aforementioned steps, culture plates were incubated at 37 °C in a humidified atmosphere of CO_2_ 5%. After 18 h, 100 μL of the solution was transferred to a 96-well plate and an optimal density reading was conducted at 595 nm. The experiments were performed in biological triplicate.

#### 3.3.3. Hemolysis Assay with or without an AC Magnetic Field

The hemocompatibility of different concentrations of hybrid nanoparticles was evaluated here by a hemolysis test, to determine the hemolytic potential of these nanoparticles. In particular, hemolysis is associated with hemoglobin release into the surrounding fluid (blood plasma or normal saline) caused by the disruption (lysis) of the erythrocytic membrane in vivo (general blood circulation) or in vitro (deionized water). The effect of the magnetic field on the percentage of hemolysis was also studied. To the best of our knowledge, this was the first time that the hemolysis test was performed under the action of a magnetic field.

The hemolysis assay was used as described by Martínez-Rodríguez et al. [43], with slight modifications. For this assay, defibrinated sheep’s blood (Dsyslab) was used. The blood was washed by centrifugation at 4500× *g* for 5 min in a clinical centrifuge and the plasma and buffer coat were removed by aspiration. The erythrocytic pellet was washed three times with 0.9% NaCl at 4 °C followed by centrifugation at 4500× *g* for 5 min. After washing, erythrocytes were then diluted in 0.9% NaCl (3:11). For the assay, 100 μL of erythrocytes was incubated with 900 μL of the mag1@poly sample. The mag1@poly sample was incubated at different concentrations, 0.5, 1.0, 2.5, and 5.0 mg mL^−1^. Erythrocytes incubated with 0.9% NaCl were used as the negative controls. As the positive controls, erythrocytes were lysed with sterile deionized water to test their hemolysis. The initial hemolysis did not exceed 2%.

A part of the tubes containing erythrocytes and the mag1@poly sample was put into the center of the electromagnetic induction heating coil with a magnetic field of 216 kHz for 30 min. After that, the samples were incubated for 30 min at 37 °C. Another part was not placed under an AC magnetic field and these samples were incubated for 6 h at 37 °C. Then, the samples were centrifuged at 4500× *g* for 5 min at 4 °C. A total volume of 100 μL of each sample was placed in a 96-well plate and absorbance was measured at 540 nm with an ELISA reader (Thermo Fisher Scientific, Waltham, MA, USA). Hemolysis was calculated with the formula:$$Hemolysis\left(\%\right)=\frac{sample\;absorbance-negative\;control}{positive\;control-negative\;control}\times 100$$

#### 3.3.4. Statistical Analysis

Statistical analysis for the cell viability assay data was performed using two-way ANOVA and for the hemolysis assay, using one-way ANOVA, both followed by Tukey’s post hoc test. For this, GraphPad Prism 6 software was used, and the reported percentage values are expressed as means ± SEM. Results with a *p* < 0.05 were considered statistically significant.

## 4. Conclusions

Magnetite nanoparticles were successfully synthesized and coated with poly(L-co-D,L lactic acid-co-trimethylene carbonate) and a poly(ethylene oxide)–poly(propylene oxide)–poly(ethylene oxide) triblock copolymer. The average diameter achieved for the synthesized magnetite was 3.2 nm as determined by TEM. The heating ability measurements indicate that the concentrations of 0.5, 1.0, 2.5, and 5 mg mL^−1^ of hybrid nanoparticles in the liquid medium were enough to achieve an increase in temperature of 6 °C in about 5 min. When the cells were subjected to an alternating current field, the absence of cytotoxicity at 24 and 48 h was only observed for the concentrations of 0.5 and 1.0 mg mL^−1^. For all concentrations tested, the hybrid nanoparticles were considered non-hemolytic according to ASTMF-756. Our results demonstrate the suitability of mag1@poly nanoparticles for potential hyperthermia applications, as only a sample concentration of 0.5 mg mL^−1^ is enough for achieving the required treatment temperature under an applied magnetic field of 216 kHz for 5 min. Moreover, the studied hybrid nanoparticles can avoid potential undesirable effects in healthy patients due to the biocompatibility of their polymeric shell and the friendly chemical composition of their ceramic core. Therefore, the hybrid and multifunctional nanoparticles produced in this work offer a very promising strategy for being potentially used in applications involving cancer treatment and in situ drug delivery.

Magnetite nanoparticles were successfully synthesized and coated with poly(L-co-D,L lactic acid-co-trimethylene carbonate) and a poly(ethylene oxide)–poly(propylene oxide)–poly(ethylene oxide) triblock copolymer. The average diameter achieved for the synthesized magnetite was 3.2 nm as determined by TEM. The heating ability measurements indicate that the concentrations of 0.5, 1.0, 2.5, and 5 mg mL^−1^ of hybrid nanoparticles in the liquid medium were enough to achieve an increase in temperature of 6 °C in about 5 min. When the cells were subjected to an alternating current field, the absence of cytotoxicity at 24 and 48 h was only observed for the concentrations of 0.5 and 1.0 mg mL^−1^. For all concentrations tested, the hybrid nanoparticles were considered non-hemolytic according to ASTMF-756. Our results demonstrate the suitability of mag1@poly nanoparticles for potential hyperthermia applications, as only a sample concentration of 0.5 mg mL^−1^ is enough for achieving the required treatment temperature under an applied magnetic field of 216 kHz for 5 min. Moreover, the studied hybrid nanoparticles can avoid potential undesirable effects in healthy patients due to the biocompatibility of their polymeric shell and the friendly chemical composition of their ceramic core. Therefore, the hybrid and multifunctional nanoparticles produced in this work offer a very promising strategy for being potentially used in applications involving cancer treatment and in situ drug delivery.

## Figures and Tables

**Figure 1 ijms-25-00493-f001:**
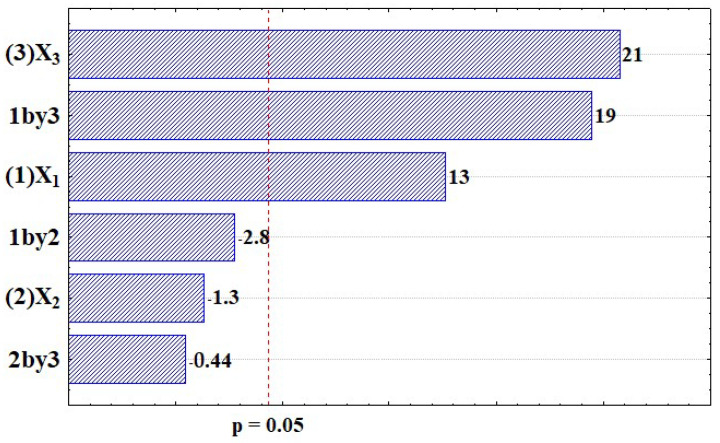
Pareto chart of standardized effects: addition time of the ammonium hydroxide solution, X_1_; mixing speed, X_2_; and temperature of the synthesis, X_3_.

**Figure 2 ijms-25-00493-f002:**
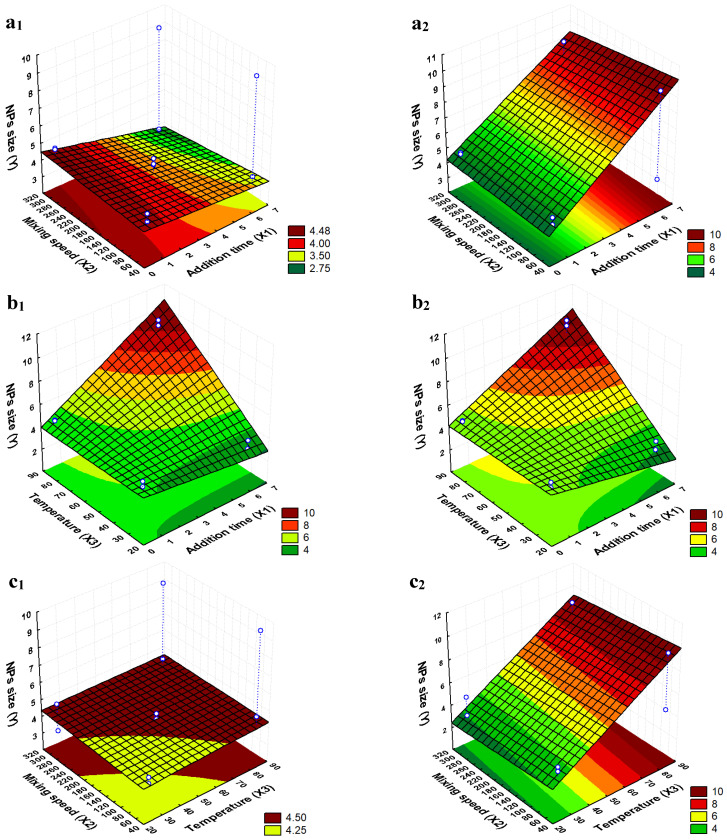
Response surfaces and contour plots for the effects of independent variables on the size of synthesized magnetite nanoparticles: (**a**) mixing speed (X_2_) and addition time of the ammonium hydroxide solution (X_1_) at 25 °C (**a1**) and 85 °C (**a2**); (**b**) temperature of the synthesis (X_3_) and addition time of the ammonium hydroxide solution (X_1_) at 50 rpm (**b1**) and 300 rpm (**b2**); and (**c**) temperature of the synthesis (X_3_) and mixing speed (X_2_) at 40 s of addition time (**c1**) and 6 min 24 s of addition time of ammonium hydroxide (**c2**).

**Figure 3 ijms-25-00493-f003:**
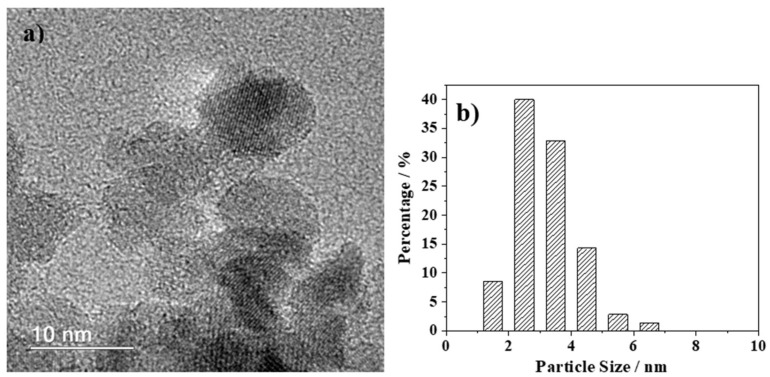
(**a**) HRTEM image and (**b**) size distribution histogram of the hybrid mag1@poly nanoparticles.

**Figure 4 ijms-25-00493-f004:**
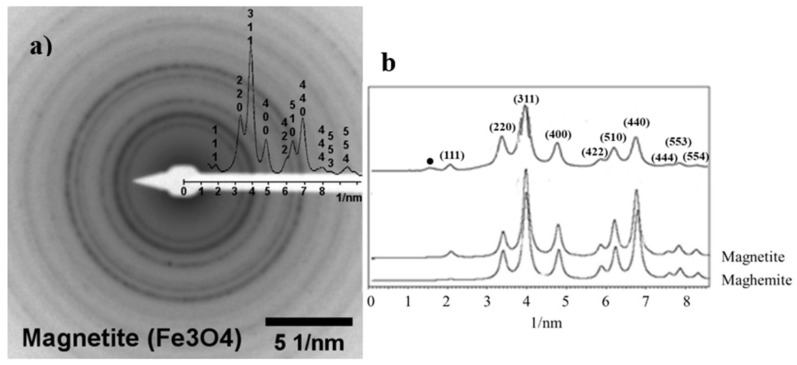
(**a**) SAD pattern for the mag1 sample with inverted contrast. (**b**) Radial profiles of the SAD pattern and simulated electron diffraction profiles for standard magnetite and maghemite (both with a centric setting space group *Fd-3 m) as a function of the reciprocal d*—spacings (d_hkl_* = d_hkl_^−1^). The intensities for all patterns were normalized in relation to that of the (311) peak reflection. The reflection marked with a black circle at 1.65 nm^−1^, corresponding to d-spacing = 0.606 nm, is ascribed to the forbidden reflection (011) that appeared due to double diffraction events.

**Figure 5 ijms-25-00493-f005:**
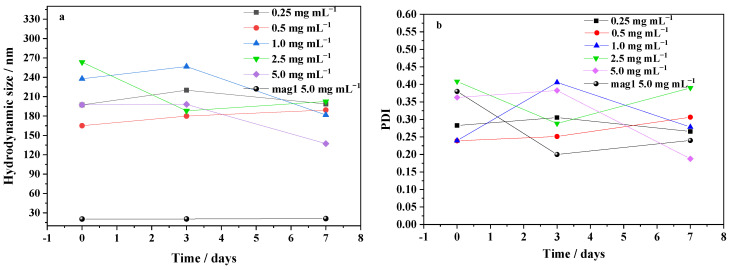
Stability of agglomerates of mag1@poly at different concentrations: (**a**) hydrodynamic size and (**b**) PDI on days 0, 3, and 7.

**Figure 6 ijms-25-00493-f006:**
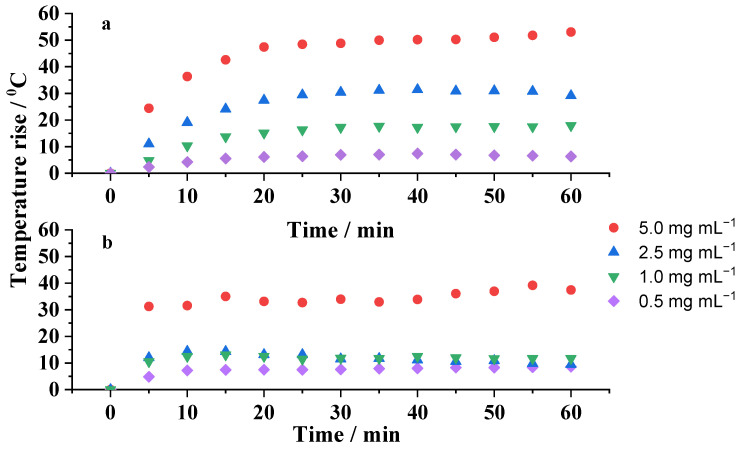
Temperature–time curves for the mag1@poly sample at different concentrations, 0.5, 1.0, 2.5, and 5.0 mg mL^−1^, under an AC field of 105 Oe and 216 kHz, dispersed in: (**a**) water and (**b**) silicone wax.

**Figure 7 ijms-25-00493-f007:**
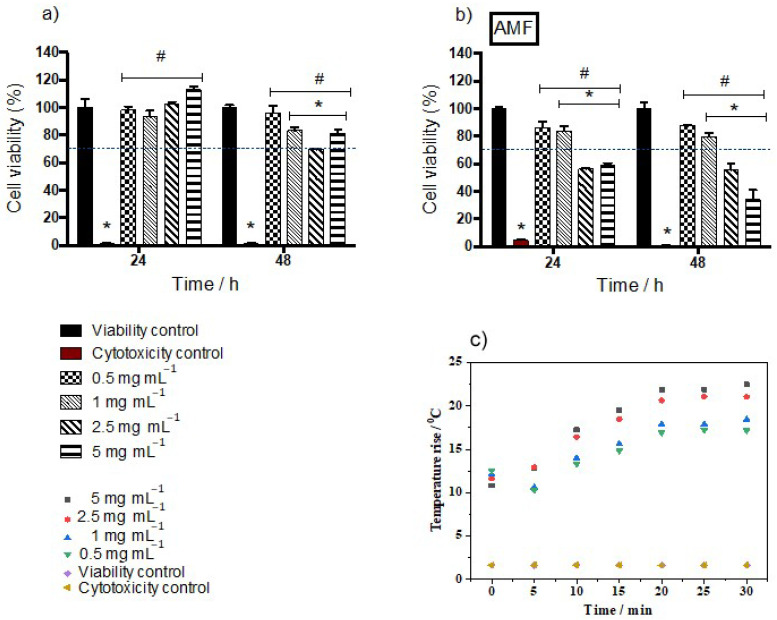
Cell viability evaluated by MTT in HUVECs exposed to four different concentrations of the mag1@poly suspension, 0.5, 1.0, 2.5, and 5 mg mL^−1^, at 24 and 48 h: (**a**) without an AC magnetic field and (**b**) with an AC magnetic field (AMF) of 105 Oe and 216 kHz for 30 min. Mean cell viability was normalized by the mean viability of the control group. The viability control used was PBS and the cytotoxicity control used was 0.5% *v*/*v* Triton™ X-100. Stars denote statistical significance at a *p* < 0.05 vs. the viability control and a # *p* < 0.05 vs. the cytotoxicity control as determined by two-way ANOVA followed by Tukey’s post hoc test. (**c**) Temperature–time curves for suspensions containing HUVEC cells exposed to the mag1@poly suspension under an AC magnetic field of 105 Oe and 216 kHz.

**Figure 8 ijms-25-00493-f008:**
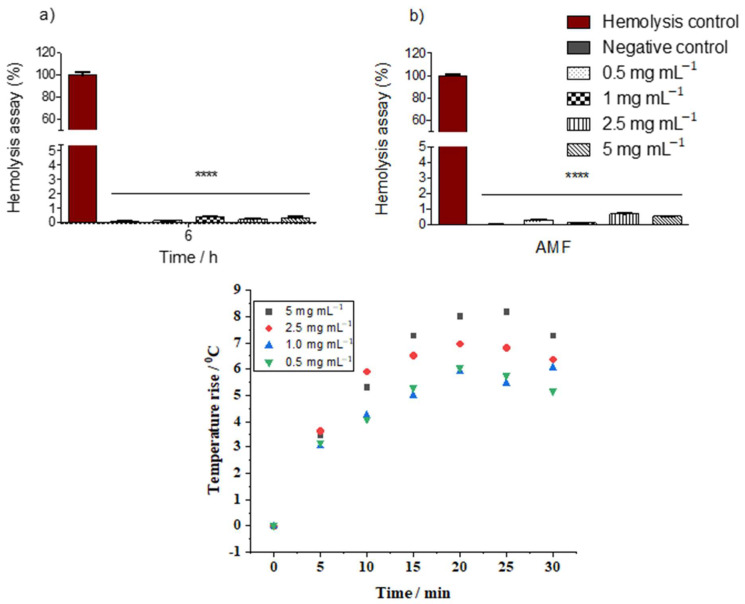
Hemolytic assay at four different concentrations of the mag1@poly suspension, 0.5, 1.0, 2.5, and 5 mg mL^−1^, at 6 h: (**a**) the percentages of hemolysis at different concentrations of the mag1@poly without an AC magnetic field and (**b**) the percentages of hemolysis at different concentrations of the mag1@poly under an AC magnetic field (AMF) of 105 Oe and 216 kHz for 30 min. The mean hemolytic assay was normalized by the mean hemolytic group. For the negative control, PBS was used and for the hemolytic control, Milli-Q water was used. Stars denote statistical significance at a *p* ≤ 0.0001 vs. the hemolytic control as determined by one-way ANOVA followed by Tukey’s post hoc test. (**c**) Temperature–time curves for suspensions containing erythrocytes exposed to different concentrations of the sample of the mag1@poly suspension at 5, 2.5, 1.0, and 0.5 mg mL^−1^ under an AC magnetic field of 105 Oe and 216 kHz.

**Table 1 ijms-25-00493-t001:** Levels of the independent factors.

Independent Factor	Level		
	Low Level (−1)	Middle Level (0)	High Level (+1)
X_1_ (time)	40 s	3 min 32 s	6 min 24 s
X_2_ (rpm)	50	175	300
X_3_ (°C)	25	55	85

**Table 2 ijms-25-00493-t002:** Factorial design 2^3^: coded levels, second-order contrast coefficients, and response.

Run	X_1_	X_2_	X_3_	X_1_X_2_	X_1_X_3_	X_2_X_3_	Y (nm)
1	−1	−1	−1	1	1	1	4.3
2	1	−1	−1	−1	−1	1	3.9
3	−1	1	−1	−1	1	−1	4.8
4	1	1	−1	1	−1	−1	3.2
5	−1	−1	1	1	−1	−1	4.8
6	1	−1	1	−1	1	−1	9.6
7	−1	1	1	−1	−1	1	4.7
8	1	1	1	1	1	1	9.2
9	0	0	0	0	0	0	4.2
10	0	0	0	0	0	0	4.6

**Table 3 ijms-25-00493-t003:** Effects of the independent factors.

Factor and Interaction	Effect Value	*p*-Value
X_1_	1.84	0.006
X_2_	−0.195	0.314
X_3_	3.03	0.002
X_1_X_2_	−0.405	0.110
X_1_X_3_	2.84	0.003
X_2_X_3_	−0.065	0.701

**Table 4 ijms-25-00493-t004:** Average physical–chemical parameters of the hybrid mag1@poly nanoparticles. Z-Ave = mean particle size and PDI = polydispersity index.

Sample	Z-Ave/nm	PDI	ZP/mV
Blank nanoparticles *	243 ± 6	0.130	−18 ± 6
mag1	78 ± 1	0.194	−22 ± 4
mag1@poly	193 ± 5	0.219	−15 ± 6

* Control shell nanoparticles (without magnetic cores).

## Data Availability

Data are contained within the article.

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
