# Peer review of "Heating Capacity and Biocompatibility of Hybrid Nanoparticles for Magnetic Hyperthermia Treatment"

_ijms, 2023, doi:10.3390/ijms25010493_

Round 1

Reviewer 1 Report

Comments and Suggestions for Authors

Gomes and colleagues synthesized and characterized iron oxide nanoparticles coated with two biocompatible polymers. The study delved into the physical attributes, heating capacity, and cytotoxicity of these particles. While the manuscript offers valuable insights, the addition of a control group—specifically uncoated iron oxide nanoparticles—could help underline the advantages of polymer usage. Below are some specific suggestions:

1. The abbreviation "SDM" appears on line 82 in the Introduction and again on line 522 in the Methods section. Could the authors clarify this abbreviation?

2. The choice of the polymers PLDLA-co-TMC and PEO-PPO-PEO merits further explanation. It would be enlightening to understand the individual benefits of each polymer and the rationale behind their concurrent application in the nanoparticle synthesis process.

3. The authors should provide justification for considering the smallest particle size as the optimal condition.

4. The term "blank nanoparticles" is used in Table 4. A clearer definition or description would be helpful.

5. Was the stability of the polymer coating for a 7-day testing period investigated at room temperature? To enhance the comparative analysis, it would be beneficial to include "mag1" at varying concentrations as controls. Additionally, considering the inclusion of PDI results alongside the size changes could provide a more comprehensive overview.

6. For a more robust cytotoxicity analysis, I suggest integrating a control using uncoated "mag1" nanoparticles in Figure 7.

7. The legends and associated descriptions for Figure 7c seem to be absent or incomplete. Could the authors elucidate on the terms "CP" and "CN"?

8. In the conclusion part, it would be enriching if the authors touched upon potential research avenues, particularly focusing on the drug-release capabilities of this nanoparticle platform.

Author Response

Reviewer 1

Gomes and colleagues synthesized and characterized iron oxide nanoparticles coated with two biocompatible polymers. The study delved into the physical attributes, heating capacity, and cytotoxicity of these particles. While the manuscript offers valuable insights, the addition of a control group—specifically uncoated iron oxide nanoparticles—could help underline the advantages of polymer usage. Below are some specific suggestions:

Reviewer comment:

  1. The abbreviation "SDM" appears on line 82 in the Introduction and again on line 522 in the Methods section. Could the authors clarify this abbreviation?

Authors´ response:

Please note that the abbreviation has been already added and defined, for the first time, in the Line 62.

Reviewer comment:

  1. The choice of the polymers PLDLA-co-TMC and PEO-PPO-PEO merits further explanation. It would be enlightening to understand the individual benefits of each polymer and the rationale behind their concurrent application in the nanoparticle synthesis process.

Authors´ response:

Thank you for the interesting remark. Accordingly, a new sentence was added in the revised version of the manuscript (Lines 80 to 97) aiming to satisfy this requirement.

Reviewer comment:

  1. The authors should provide justification for considering the smallest particle size as the optimal condition.

Authors´ response:

Thank you for the note. In order to address it, the following sentence was added in the revised version of the manuscript (Lines 121 to 127).

Reviewer comment:

  1. The term "blank nanoparticles" is used in Table 4. A clearer definition or description would be helpful.

Authors´ response:

Thank you for the remark. The clarification was added at the bottom of Table 4.  Moreover, the following sentence was added (Lines 563 – 565): “Control shell nanoparticles (without magnetic cores), defined as ‘blank nanoparticles’ at the bottom of Table 4, were prepared using the same experimental procedure.

Reviewer comment:

  1. Was the stability of the polymer coating for a 7-day testing period investigated at room temperature? To enhance the comparative analysis, it would be beneficial to include "mag1" at varying concentrations as controls. Additionally, considering the inclusion of PDI results alongside the size changes could provide a more comprehensive overview.

Authors´ response:

Thank you for the request. The following sentence was added (Lines 578-579): “The measurements were done at room temperature.” The stability data for sample mag1 were also plotted in Figure 5a, and the PDI values were included in Figure 5b.

Reviewer comment:

  1. For a more robust cytotoxicity analysis, I suggest integrating a control using uncoated "mag1" nanoparticles in Figure 7.

Authors´ response:

The authors welcome this comment. However, considering that the mag1 sample is functionalized with tetramethylammonium hydroxide, a strong base, this uncoated sample cannot be used in in vitro tests. It would kill many cells. This sample is only prepared to form a ferrofluid with the magnetic nanoparticles to be later coated it with a biocompatible material.

Reviewer comment:

  1. The legends and associated descriptions for Figure 7c seem to be absent or incomplete. Could the authors elucidate on the terms "CP" and "CN"?

Authors´ response:

Dear reviewer, the terms were modified.

Reviewer comment:

  1. In the conclusion part, it would be enriching if the authors touched upon potential research avenues, particularly focusing on the drug-release capabilities of this nanoparticle platform.

Authors´ response:

Thank you for this interesting suggestion! Accordingly, the following sentence has been added (Lines 683 – 686): “Therefore, the hybrid and multifunctional nanoparticles produced in this work offer a very promising strategy for being potentially used in applications involving cancer treatment and in situ drug delivery.”

Reviewer 2 Report

Comments and Suggestions for Authors

This is a solid development with good data presentation. I do have some doubts over some discussions in the manuscript, which I think the authors would need to address:

** Lines 303-305: The authors describe that "According to these results, the coated MNP can be potentially effective for localized hyperthermia treatment because it is only necessary to inject a small number of nanoparticles inside the body." The in vitro-to-in vivo correlation here appears to be elusive to me and should be elaborated further. What concentration is expected when these nanoparticles are used in vivo and how this would be related to the in vitro results shown in the manuscript?

** More explanations to understand Figure. 2 are needed. For multi-dimensional plots, any new elements not commonly seen in a conventional 2D plot should have specifications. For example, what does the grid mean and what are the colors representing? Besides, what do the hollow dots represent?

** I feel that more background and discussions are needed to read the TEM image shown in Figure 3a. The particles are not presenting clear peripheries and how the size distribution was derived?

Author Response

Reviewer 2

This is a solid development with good data presentation. I do have some doubts over some discussions in the manuscript, which I think the authors would need to address:

Reviewer comment:

** Lines 303-305: The authors describe that "According to these results, the coated MNP can be potentially effective for localized hyperthermia treatment because it is only necessary to inject a small number of nanoparticles inside the body." The in vitro-to-in vivo correlation here appears to be elusive to me and should be elaborated further. What concentration is expected when these nanoparticles are used in vivo and how this would be related to the in vitro results shown in the manuscript?

Authors´ response:

Thank you for your pertinent comment. Please note that one of the reasons for working with low concentrations of magnetic nanoparticles is to prevent their agglomeration and the released excessive local heat that, otherwise, would lead to burning the location in the living organism where it is placed. Furthermore, hyperthermia treatment is still new, with not many definite answers available so far. For example:

- In the review article “Principles of Magnetic Hyperthermia: A Focus on Using Multifunctional Hybrid Magnetic Nanoparticles” by Ihab M. Obaidat,Venkatesha Narayanaswamy, Sulaiman Alaabed, Sangaraju Sambasivam, and Chandu V. V. Muralee Gopi, the authors say: “The dipolar interaction is a long range interaction where the interaction energy is proportional to 1/r6, where r is the interparticle distance. Therefore, dipolar interactions between MNPs decrease strongly with increasing the interparticle distance. This means that particles with small concentrations will experience small dipolar interactions. Strong dipolar interactions are expected to have an impact on the magnetic relaxations of MNPs, and thus on their heating efficiency in the existence of an alternating magnetic field (AMF)… In clinical treatments using magnetic hyperthermia (MH), the concentrations of MNPs is 112 mg Fe/mL, while in MH experiments the concentrations used are much smaller, usually between 0.1 and 30 mg/mL [72,73,74,75,76,77,78,79,80,81,82,83]. The high concentration of MNPs in clinical treatments result in smaller interparticle distances and thus large dipolar interactions that could lead to agglomerations or aggregates. The agglomerations could have a negative influence on the heating efficiency of the MNPs due to hindered relaxation processes [31,73]. Although several theoretical and experimental studies were conducted to reveal the role of dipolar interactions on heating efficiency in MH, there is no complete understanding of this topic yet due to contradiction results that could have several sources [24,84,85,86].”

- In the review article “Recommendations for in vitro and in vivo testing of Magnetic nanoparticle hyperthermia combined with radiation therapy” by Spiridon V. Spirou, Sofia A. Costa Lima, Penelope Bouziotis, Sanja Vranješ-Djuric, Eleni K. Efthimiadou, Anna Laurenzana, Ana Isabel Barbosa, Ignacio Garcia-Alonso, Carlton Jones, Drina Jankovic, and Oliviero L. Gobbo, Nanomaterials, s 2018, 8, 306; doi:10.3390/nano8050306, the authors say: Unlike Radiation Therapy, however, which is a well-established treatment in cancer management, Hyperthermia needs to overcome several practical difficulties that have, so far, prevented it from realizing its full potential. As MNP-mediated Hyperthermia is a new field that has not matured yet. Clinical trials, however, failed to deliver the expected results, primarily due to difficulties in adequately heating the tumor. This was especially true for deep-seated tumors.

Reviewer comment:

** More explanations to understand Figure. 2 are needed. For multi-dimensional plots, any new elements not commonly seen in a conventional 2D plot should have specifications. For example, what does the grid mean and what are the colors representing? Besides, what do the hollow dots represent?

Authors´ response:

Thank you for the opportunity to provide further clarifications. With this aim, the following sentence has been added (Lines 200-204): “Figure 2 shows the response surfaces predicted by the mathematical model. The colors indicate the sizes of the nanoparticles: more the color moves towards the green region on the surface, the smaller are the nanoparticles obtained under the experimental conditions indicated on the horizontal axes. The hollow dots represent the conditions of the runs indicated in Tables 1 and 2.”

Reviewer comment:

** I feel that more background and discussions are needed to read the TEM image shown in Figure 3a. The particles are not presenting clear peripheries and how the size distribution was derived?

Authors´ response:

Thank you for the request. The following sentence was added (Lines 218-221): “The particle size distribution was performed based on the bright-field TEM images (not shown) utilizing the ImageJ software. The diameter of approximately 70 particles were manually measured to get the size distribution.”

Examples of images in which particles were counted:

...

Round 2

Reviewer 1 Report

Comments and Suggestions for Authors
  1. 1. On line 107, the sentence should be revised for grammatical accuracy: "Despite these polymers being biocompatible, they may induce the formation of acidic by-products..."

  2.  
  3. 2. In Figure 5a, it appears that the data for "mag1 at 5.0 mg/ml" is missing.

Author Response

Revisor 1

  1. On line 107, the sentence should be revised for grammatical accuracy: "Despite these polymers being biocompatible, they may induce the formation of acidic by-products..."

Thank you for the remark. the sentence was modified: “Despite these polymers being biocompatible, they are also likely to induce the for-mation of acidic byproducts that might lead to tissue inflammatory reactions and clinic failure” (Lines 108-110)

  1. In Figure 5a, it appears that the data for "mag1 at 5.0 mg/ml" is missing.

Reviewer comment:

Authors´ response:

In fact, the line for "mag1 at 5.0 mg/ml" is shown at the bottom of the figure, with a hydrodynamic size / nm under 30. Please, check this.
